# Exclusive Unlearning: Forgetting All Except What You Need

## Abstract

Large language models (LLMs) acquire diverse knowledge and abilities through pretraining, but broad and uncontrolled capabilities are not always desirable. Conventional unlearning aims to erase specific knowledge while preserving general fluency and reasoning. However, when the target task is clearly defined, we can remove all abilities that are not required to perform that task, rather than attempting to control a wide range of undesirable behaviors. For example, customer-care chatbots should only answer anticipated questions, and in education, a subject-specific model may be preferable to prevent unintended use.

In this paper, we take the opposite perspective from conventional unlearning and propose a method that preserves only the ability specified by a dataset while forgetting all other knowledge and abilities. Our approach is remarkably simple: we train the model on the target task via standard fine-tuning while simultaneously forcing the probability distribution over the model's generated texts to become uniform. This ensures that the model retains the capability required for the target task, while forgetting all other abilities.

We demonstrate that our method successfully retains specific abilities (extractive QA and mathematical QA) while forgetting all other knowledge and abilities. Furthermore, we show that our method more effectively removes all abilities except the designated one compared to a standard unlearning approach.

## 1 Introduction

Large language models (LLMs) acquire broad knowledge and abilities through pretraining (Roberts et al., 2020; Carlini et al., 2023; Morris et al., 2025; Chang et al., 2024), but these are not always desirable. Prior work has often highlighted the risks associated with retaining specific factual knowledge, which can raise ethical and moral concerns (Bender et al., 2021; Hartmann et al., 2023; Satvaty et al., 2025; Staab et al., 2024). For instance, factual knowledge memorized by LLMs may include personal information, raising privacy concerns, or confidential information, which could lead to security issues or copyright violations. Since LLMs are trained on massive corpora, it is nearly impossible to preemptively remove all such sensitive content from the training data. As a result, unlearning, which aims to selectively delete specific factual knowledge from LLMs at relatively low computational cost, has emerged as an important research direction (Liu et al., 2024a; Si et al., 2023; Yao et al., 2023; Wang et al., 2024b). In this context, it is crucial not only to erase targeted knowledge but also to preserve general reasoning and generation abilities.

However, we argue that the motivation for unlearning is not limited to removing specific factual knowledge. There are scenarios where the broad knowledge and abilities of LLMs are themselves problematic. For example, customer-care chatbots should not rely on arbitrary or incorrect knowledge but instead restrict their responses to a predefined set of anticipated questions. In education, it may be desirable to build subject-specific models that prevent unintended use by students. For instance, when used as a teaching assistant, an LLM should freely answer questions within a specific subject area but avoid responding to unrelated topics such as casual conversation or entertainment. In cases where the target task is clearly defined, it may be more natural to forget everything outside that scope rather than attempting to control a wide range of undesirable behaviors. Yet, existing work on unlearning has almost exclusively focused on erasing specific knowledge, and no prior study has pursued the motivation of erasing all abilities outside a designated one.

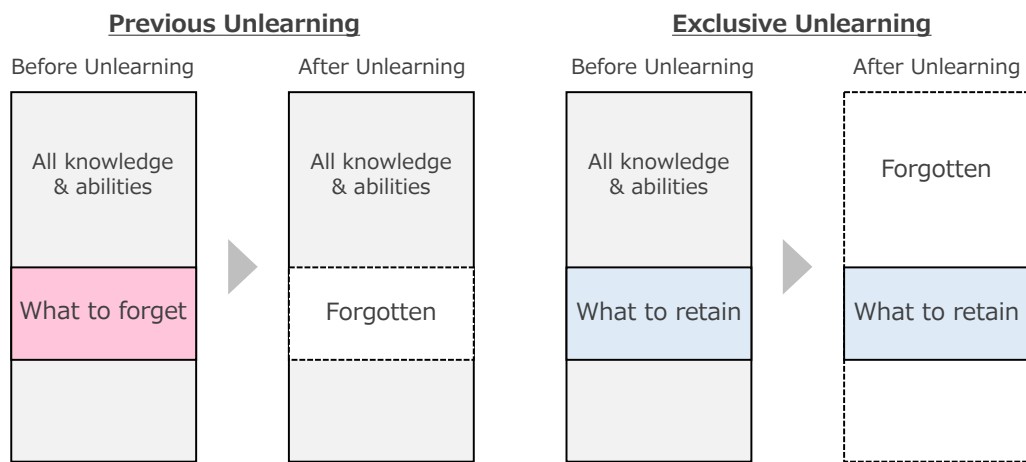

Figure 1: **Overview of Exclusive Unlearning.** Prior research aims to forget specific what to forget items while retaining the rest of a model's knowledge and abilities. In contrast, our work focuses on retaining specific what to retain abilities and forgetting everything else. When the target task is clearly defined, it is often preferable to remove all abilities not required for that task, rather than attempting to control a wide range of undesirable behaviors.

In this paper, we take the opposite perspective from conventional unlearning and propose **Exclusive Unlearning**, a method that preserves only the ability specified by a dataset while forgetting all other knowledge and abilities. For example, when preserving the ability to perform extractive question answering, the model should retain only the language skills needed to produce answers and the capability to locate and reference relevant spans in context, while forgetting all other unnecessary knowledge and abilities.

Our approach is remarkably simple. We assume that the model's generated texts encode its knowledge and abilities, and define the state in which all texts are generated with equal probability as the condition where the model has forgotten all knowledge and abilities. With this definition in place, we first sample texts from the model and then enforce a uniform-loss objective that drives the probability distribution over these generated texts toward uniformity. At the same time, we apply standard fine-tuning on the designated dataset as a retain-loss objective. This combined objective ensures that the model forgets all capabilities not defined by the retain dataset while preserving the target ability.

Through experiments, we show that when extractive QA (SQuAD (Rajpurkar et al., 2016)) and mathematical QA (GSM8K (Cobbe et al., 2021), MathQA (Amini et al., 2019)) are used as retain tasks, the model maintains these abilities at least at the same level as the pretrained model, and in some cases surpasses standard fine-tuning, while reducing performance on the comprehensive MMLU benchmark (Hendrycks et al., 2021) nearly to the chance level. Furthermore, we show that the uniform-loss objective drives the token generation probabilities toward uniformity on tasks outside the target domain, demonstrating that the model has completely forgotten knowledge and abilities unrelated to the designated task. Finally, we compare uniform loss against the commonly used negative cross entropy loss (Jang et al., 2023; Yao et al., 2024) in unlearning and show that uniform loss is superior for erasing all knowledge and abilities outside the target task.

Taken together, our results demonstrate that reversing the conventional motivation of unlearning, namely forgetting everything except the target ability, opens a new perspective for applying unlearning techniques to LLMs.

## 2 RELATED WORK

**Unlearning knowledge in Large Language Model.** The necessity of unlearning in large language models (LLMs) is increasingly recognized from both ethical and security perspectives. (Liu et al., 2024a; Si et al., 2023; Yao et al., 2023; Wang et al., 2024b) In addition, unlearning techniques are

also employed to investigate the influence of specific subsets of training data on model performance. (Isonuma & Titov, 2024; Zhao et al., 2024)

The most common approach to realizing unlearning is to retrain the LLM on the forget set using gradient ascent-based methods, while simultaneously training on an irrelevant dataset to preserve performance on unrelated tasks. This can be implemented either by adding unlearning layers (Chen & Yang, 2023) or by directly integrating the mechanism into the LLM itself (Eldan & Russinovich, 2023). Wang et al. (2024a) proposed focusing on specific spans within sequences to minimize disruption to unrelated tasks. Other strategies include in-context unlearning (Pawelczyk et al., 2024), which uses few-shot prompts to induce forgetting of particular datasets directly in the context of use, and approaches that mitigate harmful responses by collecting problematic prompts and applying techniques such as instruction tuning (Liu et al., 2024b).

Another approach applies knowledge editing techniques (Meng et al., 2023a; Mitchell et al., 2022; Huang et al., 2023; Zheng et al., 2023), in which factual statements are replaced with end-of-sequence tokens, thereby erasing knowledge from the LLM. More recently, knowledge editing has become feasible at scale (Meng et al., 2023b; Gangadhar & Stratos, 2024), and by adapting this methodology specifically for unlearning, it is becoming possible to achieve large-scale removal of specified knowledge from LLMs (Wang et al., 2025).

**Differences Between Our Approach and Related Work.** Our approach differs from existing research in its objective and method of unlearning. First, while prior work primarily aims to erase knowledge provided by a specific dataset while minimizing impact on the model's other abilities, our approach has the opposite goal: we seek to preserve only the designated ability specified by a dataset, while forgetting all other knowledge and abilities that are not required for solving that task. Thus, rather than attempting to avoid interference with other abilities, our method explicitly enforces forgetting of every ability except the one necessary for the designated dataset.

As an implementation of forgetting "all other knowledge and abilities," we adjust the model parameters so that its output distribution over self-generated text approaches a uniform distribution. To the best of our knowledge, this formulation has not appeared in prior work and constitutes a novel contribution of our approach.

From this perspective, the most suitable baselines for comparison with our method are not recent approaches that focus on preserving other abilities (Wang et al., 2024a; Meng et al., 2023b; Wang et al., 2025), but rather Gradient Ascent–based methods (Jang et al., 2023; Yao et al., 2024) that aim at large-scale forgetting.

## 3 METHODOLOGY

Our objective is to retain only the ability to solve a specific task given by a dataset, while forgetting all other knowledge and abilities.

Following the idea of conventional unlearning (Liu et al., 2024a), we adopt parameter optimization based on a combination of a forget loss and a retain loss. The overall objective function is formulated as:

$$\mathcal{L}(\theta) = \mathcal{L}_{\text{forget}}(\theta) + \lambda \cdot \mathcal{L}_{\text{retain}}(\theta), \tag{1}$$

where, in standard unlearning, the forget loss is defined on a designated forget set so that model parameters are updated in the direction of forgetting the given dataset $\mathcal{D}_{\text{forget}}$, while the retain loss ensures the preservation of the model's utility on the retain set $\mathcal{D}_{\text{retain}}$. The regularization parameter $\lambda$ controls the balance between forgetting and retaining.

The most basic setting of unlearning applies *gradient ascent*. Gradient ascent updates the model parameters to minimize the log-likelihood of $\mathcal{D}_{\text{forget}}$. In other words, we minimize the forget loss defined as follows.

$$\mathcal{L}_{\text{forget}}(\theta) = \mathbb{E}_{x \sim \mathcal{D}_{\text{forget}}} \left[ \log p_\theta(x) \right]. \tag{2}$$

However, while gradient ascent is suitable for forgetting specific knowledge, in the case of forgetting all knowledge and abilities, it may bias the distribution toward specific incorrect outputs.

Therefore, in our approach, we interpret forgetting as forcing the model's probability distribution of a token succeeding previous tokens $p_\theta(\cdot|x_{<t})$ to be uniform. This design is based on the intuition

that the model's self-generated outputs contain all of its internal knowledge and abilities. To achieve forgetting of all knowledge and abilities, we prepare $\mathcal{D}_{\text{forget}}$ by sampling texts from the model itself in advance. Specifically, we define the forget loss as the cross-entropy (CE) between the uniform distribution $p_u(\cdot)$ and the model's generation probability $p_\theta(\cdot|x_{<t})$:

$$\mathcal{L}_{\text{forget}}(\theta) = \mathbb{E}_{x \sim \mathcal{D}_{\text{forget}}} \left[ \frac{1}{T} \sum_{t=1}^{T} \text{CE}[p_u(\cdot), p_\theta(\cdot|x_{<t})] \right], \tag{3}$$

where $T$ denotes the sequence length of text $x$ and $x_{<t}$ is the sequence of the tokens generated before the time step $t$.

Furthermore, an advantage of the uniform loss is that it allows us to quantitatively assess the degree of forgetting. Specifically, when the generation probability over the entire vocabulary $V$ becomes uniform, we have $p_\theta(x|x_{<t}) = 1/|V|$ for any token $x$. In this case, the loss given by Eq. 3 evaluates to $\ln(|V|)$. Thus, our method provides a theoretical reference value that indicates whether forgetting has been successfully achieved. This helps quickly determine whether training is proceeding successfully.

For the retaining term, we follow standard fine-tuning by minimizing negative log-likelihood on $\mathcal{D}_{\text{retain}}$, preserving the designated ability:

$$\mathcal{L}_{\text{retain}}(\theta) = \mathbb{E}_{x \sim \mathcal{D}_{\text{retain}}} \left[ -\log p_\theta(x) \right]. \tag{4}$$

In summary, the model parameters are optimized to minimize the loss function in Eq. 1, which balances the forgetting and retaining objectives.

## 4 EXPERIMENTS

### 4.1 EXPERIMENTAL SETUP

To demonstrate the effectiveness of our method, we consider two settings for the forget term.

**(1) Uni CE:** Our proposed method that moves the output distribution closer to a uniform distribution. The forgetting term follows Eq. 3, the retaining term follows Eq. 4, and we minimize Eq. 1.

**(2) Neg CE:** The most basic gradient ascent setting commonly used in unlearning. The forgetting term follows Eq. 2, the retaining term follows Eq. 4, and we minimize Eq. 1.

For the forgetting dataset, we use self-generated texts from the model itself. In total, 32,000 texts are sampled (2000 steps × batch size of 16), and each text consists of 32 tokens. All generations are produced with a fixed random seed and temperature 2.0 to ensure reproducibility while encouraging diversity. The validity of this dataset choice is examined later in § 5.1.

As retaining datasets, we use the standard extractive QA dataset SQuAD (Rajpurkar et al., 2016), and the mathematical QA datasets GSM8K (Cobbe et al., 2021) and MathQA (Amini et al., 2019). These datasets allow us to demonstrate that our method can retain specific abilities in basic tasks.

To show that our method is effective across different model sizes, we conduct experiments on models ranging from approximately 1B to 7B parameters: OLMo2-1B, OLMo2-7B (OLMo et al., 2025), pythia-1.4b, pythia-6.9b (Biderman et al., 2023), GPT2-XL (1.5B), and GPT-J-6B. The rationale for selecting these models is twofold: (1) the OLMo2 and Pythia families are preferable for future research since their pretraining corpora are publicly available, and (2) GPT2-XL and GPT-J-6B are widely used in existing unlearning research and thus serve as standard baselines.

Evaluation is conducted from two perspectives: whether the model can forget all knowledge and abilities, and whether it can retain the abilities designated in the retaining datasets.

For evaluating forgetting, **we conduct both loss-based comparison using the theoretical value described in § 3 and task-based evaluation**.

**For task-based evaluation**, we use MMLU (Hendrycks et al., 2021), a comprehensive QA dataset that covers a wide range of knowledge and abilities. After training, we expect the average score across all MMLU tasks to converge near the chance rate of 0.25. MMLU is evaluated using the

Table 1: The results compare the performance of our design (Ours), which preserves only the ability given by the extractive QA (SQuAD) dataset while forgetting all others, with the original model (Base) and a model fine-tuned only on the retain objective (FT). Retention is evaluated by whether the performance on the SQuAD test set is maintained at a level comparable to the baselines, while forgetting is evaluated by whether the average accuracy on MMLU, a QA dataset covering broad knowledge and abilities, decreases to the chance rate (0.25 for four-choice questions). In our proposed setting, using cross-entropy with a uniform distribution as the forget loss is denoted as Uni-CE, and using negative cross-entropy loss is denoted as Neg-CE.

| | Retain: SQuAD↑ | | | Forget: MMLU↓ | | |
|---|---|---|---|---|---|---|
| | **Base** | **FT** | **Ours** | **Base** | **FT** | **Ours** |
| **OLMo2-1B (Uni CE)** | 0.7218 | 0.8398 | **0.8395** | 0.3653 | 0.3567 | **0.2503** |
| **OLMo2-1B (Neg CE)** | 0.7218 | 0.8398 | **0.8003** | 0.3653 | 0.3567 | **0.2269** |
| **OLMo2-7B (Uni CE)** | 0.8170 | 0.8619 | **0.8753** | 0.4437 | 0.4307 | **0.2283** |
| **OLMo2-7B (Neg CE)** | 0.8170 | 0.8619 | **0.8365** | 0.4437 | 0.4307 | **0.2267** |
| **Pythia-1.4b (Uni CE)** | 0.4879 | 0.7973 | **0.8328** | 0.3001 | 0.2930 | **0.2260** |
| **Pythia-1.4b (Neg CE)** | 0.4879 | 0.7973 | **0.7979** | 0.3001 | 0.2930 | **0.2268** |
| **Pythia-6.9b (Uni CE)** | 0.5556 | 0.8170 | **0.8334** | 0.3297 | 0.3262 | **0.2325** |
| **Pythia-6.9b (Neg CE)** | 0.5556 | 0.8170 | **0.6508** | 0.3297 | 0.3262 | **0.2234** |
| **GPT2-XL (Uni CE)** | 0.5030 | 0.7637 | **0.0020** | 0.2938 | 0.2891 | **0.2270** |
| **GPT2-XL (Neg CE)** | 0.5030 | 0.7637 | **0.8365** | 0.2938 | 0.2891 | **0.2270** |
| **GPT-J-6B (Uni CE)** | 0.5791 | 0.8351 | **0.8596** | 0.3402 | 0.3491 | **0.2485** |
| **GPT-J-6B (Neg CE)** | 0.5791 | 0.8351 | **0.7463** | 0.3402 | 0.3491 | **0.2289** |

MMLU-continuation setting of lm-evaluation-harness (Gao et al., 2023), which computes average accuracy based on log-likelihood over four choices.

**For loss-based comparison**, the forget loss is evaluated by preparing 200 sentences generated in advance by the model itself (distinct from the training data) as the forgetting evaluation set. We then measure the following two quantities and assess whether (1) and (2) approach the target value $\ln(|V|)$, where $|V|$ denotes the model's vocabulary size:

**(1) Train Forget Loss:** Cross-entropy loss with the uniform distribution on the forgetting training data at the final training step.

**(2) Eval Forget Loss:** Cross-entropy loss with the uniform distribution on the forgetting evaluation set after training.

Retain loss is evaluated by comparing the loss on the test retaining datasets between our method and the setting with only the retaining term, in order to verify whether our approach preserves abilities to a degree comparable to standard fine-tuning.

For retaining evaluation, we measure performance on the test sets of each retaining dataset using lm-evaluation-harness: For SQuAD, we evaluate using average accuracy under the contain metric, which checks whether the gold answer appears in the generated text. For GSM8K, we report average accuracy with the Exact Match metric. For MathQA, we use average accuracy based on log-likelihood classification over five choices.

For retaining evaluation, we compare performance under three different settings:

**(1) Base:** The original pretrained model before any additional training, which provides a reference for the model's initial ability.

**(2) FT:** A model trained only with the retain term, corresponding to standard fine-tuning on the retain datasets. This shows the performance when solely preserving the designated ability without forgetting.

Table 2: The results compare our design (Ours), which preserves only the ability from the Mathematical QA (GSM8K) dataset while forgetting all others, with the original model (Base) and a model fine-tuned only on the retain objective (FT). Retention is measured by whether performance on the GSM8K test set matches the baselines, and forgetting by whether the average accuracy on MMLU, a broad QA dataset, drops to the chance rate of 0.25 (four-choice). In our setting, cross-entropy with a uniform distribution is denoted Uni-CE, and negative cross-entropy is denoted Neg-CE.

| | Retain: GSM8K↑ | | | Forget: MMLU↓ | | |
|---|---|---|---|---|---|---|
| | **Base** | **FT** | **Ours** | **Base** | **FT** | **Ours** |
| **OLMo2-1B (Uni CE)** | 0.3351 | 0.3472 | **0.3230** | 0.3653 | 0.3440 | **0.2339** |
| **OLMo2-1B (Neg CE)** | 0.3351 | 0.3472 | **0.2472** | 0.3653 | 0.3440 | **0.2314** |
| **OLMo2-7B (Uni CE)** | 0.6808 | 0.6808 | **0.6679** | 0.4437 | 0.4265 | **0.2413** |
| **OLMo2-7B (Neg CE)** | 0.6808 | 0.6808 | **0.4541** | 0.4437 | 0.4265 | **0.2243** |
| **Pythia-1.4b (Uni CE)** | 0.0121 | 0.0788 | **0.0766** | 0.3001 | 0.2755 | **0.2254** |
| **Pythia-1.4b (Neg CE)** | 0.0121 | 0.0788 | **0.0265** | 0.3001 | 0.2755 | **0.2295** |
| **Pythia-6.9b (Uni CE)** | 0.0250 | 0.0561 | **0.0660** | 0.3297 | 0.3032 | **0.2213** |
| **Pythia-6.9b (Neg CE)** | 0.0250 | 0.0561 | **0.0000** | 0.3297 | 0.3032 | **0.2218** |
| **GPT2-XL (Uni CE)** | 0.0121 | 0.0243 | **0.0000** | 0.2938 | 0.2745 | **0.2260** |
| **GPT2-XL (Neg CE)** | 0.0121 | 0.0243 | **0.0000** | 0.2938 | 0.2745 | **0.2267** |
| **GPT-J-6B (Uni CE)** | 0.0349 | 0.0796 | **0.0864** | 0.3402 | 0.3220 | **0.2485** |
| **GPT-J-6B (Neg CE)** | 0.0349 | 0.0796 | **0.0000** | 0.3402 | 0.3220 | **0.2290** |

**(3) Ours:** Our proposed method, which combines the retaining term with the forget term, aiming to preserve only the designated ability while forgetting all others. The retaining performance is expected to be at least comparable to (1) and ideally close to (2).

As for implementation details, experiments with models smaller than 1.5B parameters (OLMo2-1B, pythia-1.4b) are conducted on 8 × A100-40GB GPUs, while experiments with models larger than 1.5B (OLMo2-7B, pythia-6.9b, GPT2-XL, GPT-J-6B) are conducted on 8 × H200-141GB GPUs. For numerical precision, OLMo2-7B, pythia-6.9b, and GPT-J-6B are trained in bfloat16, while the other models use float32. Unless otherwise noted, the balance parameter $\lambda$ between the retaining and forgetting terms is set to 1.0. An analysis of hyperparameter sensitivity is presented in § 5.2.

## 4.2 TASK BASED EVALUATION (PERFORMANCE ON EXTRACTIVE QA)

We first evaluate our method on the SQuAD dataset, preserving only the ability to extract answers from context. The results are shown in Tbl. 1. For many models except GPT2-XL, SQuAD accuracy is not only maintained at the level of the pre-trained model (Base) but often improved, while accuracy on MMLU, which tests broad knowledge and abilities, drops to about the chance rate of 0.25. These results demonstrate that our objective of preserving a specific ability while forgetting all others can be achieved on SQuAD. We also show that generating sentences from the model itself and using them as forgetting data effectively erases broad knowledge and abilities.

In addition, compared to the conventional negative CE loss, our uniform CE formulation consistently performs better across most models. The advantage is especially pronounced for pythia-6.9b and GPT-J-6B, highlighting the superiority of uniform forgetting.

## 4.3 TASK BASED EVALUATION (PERFORMANCE ON MATHEMATICAL QA)

Next, we evaluate the performance of our method using the GSM8K and MathQA datasets under the setting where only the ability to solve the mathematical tasks given by the retain dataset is preserved. The results with GSM8K as the retain dataset are shown in Tbl. 2, and those with MathQA are shown in Tbl. 3.

Table 3: The results compare our design (Ours), which preserves only the ability from the Mathematical QA (MathQA) dataset while forgetting all others, with the original model (Base) and a model fine-tuned only on the retain objective (FT). Retention is measured by whether performance on the MathQA test set matches the baselines, and forgetting by whether the average accuracy on MMLU, a broad QA dataset, drops to the chance rate of 0.25 (four-choice). In our setting, cross-entropy with a uniform distribution is denoted Uni-CE, and negative cross-entropy is denoted Neg-CE.

| | Retain: MathQA↑ | | | Forget: MMLU↓ | | |
|---|---|---|---|---|---|---|
| | Base | FT | Ours | Base | FT | Ours |
| OLMo2-1B (Uni CE) | 0.2988 | 0.3454 | **0.3451** | 0.3653 | 0.3544 | **0.2299** |
| OLMo2-1B (Neg CE) | 0.2988 | 0.3454 | **0.2831** | 0.3653 | 0.3544 | **0.2285** |
| OLMo2-7B (Uni CE) | 0.3809 | 0.4164 | **0.4104** | 0.4437 | 0.4393 | **0.2327** |
| OLMo2-7B (Neg CE) | 0.3809 | 0.4164 | **0.3430** | 0.4437 | 0.4393 | **0.2248** |
| Pythia-1.4b (Uni CE) | 0.2523 | 0.2787 | **0.2797** | 0.3001 | 0.2925 | **0.2285** |
| Pythia-1.4b (Neg CE) | 0.2523 | 0.2787 | **0.2536** | 0.3001 | 0.2925 | **0.2281** |
| Pythia-6.9b (Uni CE) | 0.2583 | 0.2841 | **0.2814** | 0.3297 | 0.3255 | **0.2240** |
| Pythia-6.9b (Neg CE) | 0.2583 | 0.2841 | **0.2472** | 0.3297 | 0.3255 | **0.2239** |
| GPT2-XL (Uni CE) | 0.2358 | 0.2381 | **0.2044** | 0.2938 | 0.2967 | **0.2243** |
| GPT2-XL (Neg CE) | 0.2358 | 0.2381 | **0.2054** | 0.2938 | 0.2967 | **0.2302** |
| GPT-J-6B (Uni CE) | 0.2667 | 0.2968 | **0.2941** | 0.3402 | 0.3448 | **0.2331** |
| GPT-J-6B (Neg CE) | 0.2667 | 0.2968 | **0.2482** | 0.3402 | 0.3448 | **0.2251** |

Table 4: Loss-based evaluation of forgetting and retaining with our method. Train Forget Loss is defined as the cross-entropy loss with the uniform distribution on the training data after training. Eval Forget Loss is computed using test data generated in advance by the model itself, by measuring their cross-entropy loss after training. Theoretically, these values should converge to $\ln(|V|)$, where $|V|$ is the vocabulary size, thereby allowing us to quantify the degree of forgetting. Retain loss is defined as the cross-entropy on the retention evaluation datasets, where "our" denotes our proposed method and "FT" denotes standard fine-tuning with only the retention term.

| | Forget | | | Retain | |
|---|---|---|---|---|---|
| | Train Loss | Eval Loss | Theoretical Loss | Ours↓ | FT↓ |
| OLMo2-1B | 11.52 | 11.42 | 11.52 | 0.7312 | 0.7806 |
| OLMo2-7B | 11.53 | 11.74 | 11.52 | 0.3313 | 0.2919 |
| Pythia-1.4b | 10.83 | 10.70 | 10.83 | 0.8576 | 0.8987 |
| Pythia-6.9b | 10.85 | 10.58 | 10.83 | 0.7712 | 0.8131 |
| GPT2-XL | 10.83 | 10.77 | 10.82 | 0.8866 | 0.9175 |
| GPT-J-6B | 10.85 | 10.68 | 10.83 | 0.7530 | 0.7953 |

For GSM8K, our method with the uniform-distribution loss achieves performance at least comparable to the pre-trained model (Base) for all models except GPT2-XL, and both pythia-6.9b and GPT-J-6B even surpass standard fine-tuning (FT) in accuracy. At the same time, the average task accuracy on MMLU, which tests broad knowledge and abilities, is reduced to near the chance rate.

Similarly, on MathQA, all models except GPT2-XL maintain accuracy comparable to FT, while the MMLU accuracy drops to around 0.25. These results demonstrate that our objective of preserving only the targeted ability while forgetting all others can also be achieved in mathematical tasks.

In contrast, with the conventional negative CE loss, which has been used in the context of forgetting specific abilities while retaining others, both GSM8K and MathQA show weaker retention of the targeted ability compared to our uniform-distribution loss method.

Table 5: Loss-based evaluation of forgetting with different sources for the forgetting dataset, under the setting where MathQA is used as the retain dataset. "gen" denotes sentences sampled from the model's own generations, "wiki" denotes sentences sampled from the Wikipedia subset of OLMo's pretraining corpus, and "pile" denotes sentences sampled from the Pile dataset used in Pythia pre-training. The evaluation columns indicate whether forgetting is measured on generations (eval gen) or on the pretraining corpus (eval wiki/pile). The last column shows the theoretical target loss $\ln(|V|)$ determined by the model vocabulary size.

|  | Forget Loss | | |
| --- | --- | --- | --- |
|  | **eval gen** | **eval wiki/pile** | **Target Loss** |
| **OLMo2-1B (train gen)** | 11.40 | 11.41 | 11.52 |
| **OLMo2-1B (train wiki)** | 11.16 | 11.46 | 11.52 |
| **pythia-1.4b (train gen)** | 10.74 | 10.79 | 10.83 |
| **pythia-1.4b (train pile)** | 10.73 | 10.80 | 10.83 |

### 4.4 LOSS-BASED EVALUATION

We conducted a loss-based evaluation of both forgetting and retention using our proposed method. Here, we report results under the setting where SQuAD is used as the retaining dataset. We show results of GSM8k and MathQA in App. B

As shown in Tbl. 4, for the forget loss, both Train Forget Loss and Eval Forget Loss are close to the theoretical forget loss determined by the vocabulary size across all models, demonstrating that our method successfully achieves uniform forgetting of the model's knowledge and abilities. For retain loss, our method achieves losses comparable to those of standard fine-tuning with only the retaining term across all models, indicating that both forgetting and retention are simultaneously achieved in terms of loss. Notably, OLMo2-1B, Pythia-1.4b, GPT2-XL, and GPT-J-6B achieve even lower retain loss than standard fine-tuning.

GPT2-XL, which failed to retain abilities in task-based evaluation, still achieved lower retain loss compared to standard fine-tuning, suggesting that some structural limitation may specifically affect its ability to produce correct answers despite showing lower loss.

## 5 DISCUSSION

### 5.1 VERIFICATION OF OUR FORGETTING STRATEGY

To validate the effectiveness of our forgetting-set sampling approach, we compare it with settings where forgetting data are randomly drawn from pretraining corpora, which are publicly available for OLMo2-1B and Pythia-1.4b, while using MathQA as the retain dataset. The results on the other datasets are presented in App. C.

For OLMo2-1B, whose pretraining corpus (olmo-mix-1124 (OLMo et al., 2025)) consists of multiple sources, we use the wiki subset as a representative corpus and compare our method with forgetting trained on Wikipedia.

For Pythia-1.4b, pretrained entirely on the Pile (Gao et al., 2020), we randomly sample from the whole corpus and compare our method with forgetting on the full Pile.

As in previous sections, forgetting is evaluated using the theoretical value of the forget loss. Results are summarized in Tbl. 5, where rows denote the model and forgetting data source, and columns the evaluation data. For example, "train = wiki, eval = gen" for OLMo2-1B means forgetting training used wiki samples, while evaluation used model generations.

The results show that for OLMo2-1B, our method (training on model generations) yields evaluation losses on both "gen" and "wiki" close to the theoretical value (11.52), whereas training on wiki causes larger deviations on "gen", indicating residual knowledge.

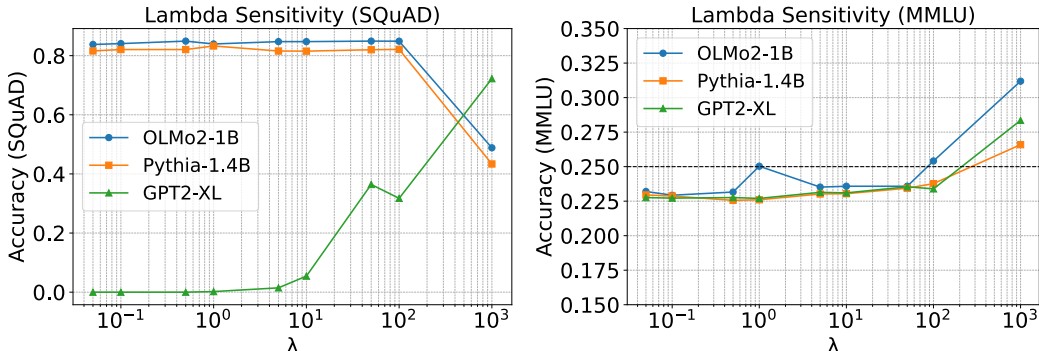

Figure 2: Hyperparameter Sensitivity of the Balance Parameter $\lambda$. The horizontal axis represents the parameter $\lambda$ as defined in Eq. 1, and the vertical axis shows task accuracy on the retain task SQuAD (left) and the forgetting evaluation task MMLU (right).

For Pythia-1.4b, random sampling from the model and from the Pile give comparable losses, showing our method matches corpus-based forgetting. Importantly, most LMs lack public pretraining corpora, and large corpora make random sampling impractical. Thus, our sampling method provides a more scalable strategy for erasing knowledge and abilities.

### 5.2 HYPERPARAMETER SENSITIVITY OF THE BALANCE PARAMETER

We investigated the hyperparameter sensitivity of the balance parameter $\lambda$, which is crucial for this experiment, under the setting of retaining the ability on SQuAD while forgetting all other knowledge and abilities. The models examined were OLMo2-1B, Pythia-1.4B, and GPT2-XL, for which our method did not generally perform well. We conducted experiments with $\lambda \in \{0.05, 0.10, 0.50, 1.00, 5.00, 10.00, 50.00, 100.00, 1000.00\}$.

For OLMo2-1B and Pythia-1.4B, we found that across a wide range of $\lambda$ values from 0.05 to 100.00, the models were able to retain SQuAD performance above 80% (comparable to fine-tuning with only the retain term, as shown in Table 2), while simultaneously reducing MMLU accuracy to near the chance rate. This indicates that our method is generally stable with respect to the balance parameter.

In contrast, for GPT2-XL, although increasing the weight of the retain term led to substantial improvement in SQuAD performance (particularly for $\lambda$ values between 100 and 1000), the MMLU accuracy also increased at the same time. This suggests that it is difficult to achieve both forgetting and retaining simultaneously in this case. Among the models we tested, GPT2-XL was the only one showing such behavior, but it highlights an limitation: for certain models, our method may fail to achieve the desired balance between forgetting and retaining.

## 6 CONCLUSION

In this work, we introduced Exclusive Unlearning, a new perspective on unlearning in LLMs. Unlike conventional approaches that erase specific knowledge while preserving broad capabilities, our method preserves only the ability specified by a target dataset and forgets everything else.

Our method is simple and effective. By applying a uniform-loss objective to self-generated texts and fine-tuning on the retain dataset, we enforce forgetting of all knowledge outside the target scope. Across diverse model families and scales, we show that Exclusive Unlearning retains designated abilities in extractive and mathematical QA, while reducing performance on broad benchmarks such as MMLU to near chance.

Overall, Exclusive Unlearning provides a practical, theoretically grounded framework for selectively constraining LLMs. By inverting the conventional unlearning objective, it enables models to be systematically tailored for focused, well-defined purposes, such as customer-facing chatbots or subject-specific educational tools.

ETHICS STATEMENT

Throughout this study, we have adhered to the ICLR Code of Ethics. We conducted our research with integrity, respecting all applicable laws and ethical standards, and carefully considered the broader societal implications of our work.

REPRODUCIBILITY STATEMENT

We provide a clear experimental setup in § 4.1. We provide our code and data as supplementary material to ensure the reproducibility.

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

Table 6: Loss-based evaluation of forgetting and retaining with our method. (retain data is **GSM8K**)

| | Forget | | | Retain | |
|---|---|---|---|---|---|
| | **Train Loss** | **Eval Loss** | **Theoretical Loss** | **Ours↓** | **FT↓** |
| **OLMo2-1B** | 11.52 | 11.42 | 11.52 | 0.3776 | 0.3784 |
| **OLMo2-7B** | 11.50 | 11.73 | 11.52 | 0.3886 | 0.3951 |
| **Pythia-1.4b** | 10.83 | 10.70 | 10.83 | 0.3733 | 0.3549 |
| **Pythia-6.9b** | 10.81 | 10.49 | 10.83 | 0.3782 | 0.3572 |
| **GPT2-XL** | 10.83 | 10.77 | 10.82 | 0.4002 | 0.3784 |
| **GPT-J-6B** | 10.81 | 10.58 | 10.83 | 0.3621 | 0.3450 |

Table 7: Loss-based evaluation of forgetting and retaining with our method. (retain data is **MathQA**)

| | Forget | | | Retain | |
|---|---|---|---|---|---|
| | **Train Loss** | **Eval Loss** | **Theoretical Loss** | **Ours↓** | **FT↓** |
| **OLMo2-1B** | 11.52 | 11.40 | 11.52 | 0.1318 | 0.1482 |
| **OLMo2-7B** | 11.50 | 11.63 | 11.52 | 0.1119 | 0.1439 |
| **Pythia-1.4b** | 10.83 | 10.73 | 10.83 | 0.1433 | 0.1582 |
| **Pythia-6.9b** | 10.81 | 10.60 | 10.83 | 0.1818 | 0.1871 |
| **GPT2-XL** | 10.83 | 10.78 | 10.82 | 0.1705 | 0.1797 |
| **GPT-J-6B** | 10.81 | 10.67 | 10.83 | 0.1724 | 0.1799 |

Yang Zhao, Li Du, Xiao Ding, Kai Xiong, Zhouhao Sun, Shi Jun, Ting Liu, and Bing Qin. Deciphering the impact of pretraining data on large language models through machine unlearning. In Lun-Wei Ku, Andre Martins, and Vivek Srikumar (eds.), *Findings of the Association for Computational Linguistics: ACL 2024*, pp. 9386–9406, Bangkok, Thailand, August 2024. Association for Computational Linguistics. doi: 10.18653/v1/2024.findings-acl.559. URL https://aclanthology.org/2024.findings-acl.559/.

Ce Zheng, Lei Li, Qingxiu Dong, Yuxuan Fan, Zhiyong Wu, Jingjing Xu, and Baobao Chang. Can we edit factual knowledge by in-context learning?, 2023. URL https://arxiv.org/abs/2305.12740.

# A  THE USE OF LARGE LANGUAGE MODELS (LLMs)

In the development of code and the proofreading and revision of this paper, we made use of AI assistants, including large language models. All code snippets and textual content generated with the assistance of these tools were carefully reviewed and revised by the authors to ensure scientific accuracy, reliability, and ethical compliance.

# B  DETAILS OF LOSS-BASED EVALUATION

In § 4.4, we presented a loss-based evaluation of our method for forgetting all knowledge and abilities while retaining specific ones, and reported the results when SQuAD was used as the retaining dataset. In this section, we provide further details by showing the results when GSM8K and MathQA are used as retaining datasets, reported in Tbl. 6 and Tbl. 7, respectively.

As with the case where SQuAD served as the retaining dataset, both Train and Eval Forget loss approach Teoretical Forget Loss, while the retain loss are comparable to those obtained by standard fine-tuning on the respective retaining datasets. These results demonstrate that our method achieves loss-based forgetting of knowledge and abilities outside the target dataset, while successfully retaining the abilities associated with the designated dataset.

Table 8: Loss-based evaluation of forgetting with different sources for the forgetting dataset, under the setting where SQuAD is used as the retention dataset.

| | Forget Loss | | |
| --- | --- | --- | --- |
| | eval gen | eval wiki/pile | Target Loss |
| **OLMo2-1B (train gen)** | 11.42 | 11.25 | 11.52 |
| **OLMo2-1B (train wiki)** | 11.26 | 11.38 | 11.52 |
| **pythia-1.4b (train gen)** | 10.70 | 10.72 | 10.83 |
| **pythia-1.4b (train pile)** | 10.69 | 10.75 | 10.83 |

Table 9: Loss-based evaluation of forgetting with different sources for the forgetting dataset, under the setting where GSM8K is used as the retention dataset.

| | Forget Loss | | |
| --- | --- | --- | --- |
| | eval gen | eval wiki/pile | Target Loss |
| **OLMo2-1B (train gen)** | 11.42 | 11.37 | 11.52 |
| **OLMo2-1B (train wiki)** | 11.07 | 11.43 | 11.52 |
| **pythia-1.4b (train gen)** | 10.70 | 10.78 | 10.83 |
| **pythia-1.4b (train pile)** | 10.72 | 10.79 | 10.83 |

## C   DETAILS OF VERIFICATION OF OUR FORGETTING STRATEGY

In § 5.1, we compared our proposed approach, which is sampling sentences from the model itself for forgetting training, with two alternatives: random sampling from a large-scale corpus within the pretraining data (wiki) and random sampling from the entire pretraining corpus (Pile). The results showed that our method achieves superior forgetting performance compared to wiki sampling, and performance equivalent to sampling from the full pretraining corpus, while being more efficient in terms of storage and computation cost.

In the main text, we presented results using MathQA as the retention dataset. Additional results for SQuAD and GSM8K are reported in Tbl. 8 and Tbl. 9. For GSM8K, the same tendency as MathQA is observed. For SQuAD, however, when using the "gen" setting (training on model generations), the evaluation loss on wiki samples deviates further from the theoretical target than the evaluation loss on generations, which might appear as insufficient forgetting. This is explained by the fact that SQuAD is constructed from Wikipedia: due to the retention term, training on generated texts leads to slightly lower evaluation loss on wiki samples. Thus, this behavior is consistent and expected rather than a failure of forgetting.

