# OpenReview forum: "Exclusive Unlearning: Forgetting All Except What You Need"
_ICLR.cc/2026/Conference — Submitted to ICLR 2026_

### Official Review · Reviewer_RGD1 · 2025-10-30

**Soundness:** 1
**Presentation:** 1
**Contribution:** 1
**Rating:** 2
**Confidence:** 4

**Summary:**

This paper introduces Exclusive Unlearning, a novel perspective on model unlearning for large language models (LLMs). Unlike conventional unlearning, which aims to remove specific knowledge while maintaining overall fluency and reasoning ability, this work proposes to retain only the capabilities specified by a target dataset and forget everything else. The proposed method jointly applies a standard fine-tuning objective on the retain dataset (e.g., SQuAD, GSM8K) and a “uniform-loss” objective that drives the output token distribution toward uniformity, which is claimed to erase all other knowledge and abilities. Experiments on various models (OLMo, Pythia, GPT2-XL, GPT-J) show that their method can preserve the target task ability while reducing general knowledge (MMLU) performance to chance level.

**Strengths:**

The paper provides a clear conceptual contrast to traditional unlearning — instead of “forgetting a part,” it attempts to “forget everything except one capability,” which is an interesting inversion of perspective.

The proposed approach is technically simple (a combination of standard fine-tuning and uniform-distribution regularization) and easy to implement.

The experimental setup is well-structured, covering multiple model families and tasks (extractive QA and math QA).

The loss-based evaluation introduces a theoretically interpretable measure (approaching ln|V|) for quantifying the degree of forgetting, which is a neat diagnostic tool.

**Weaknesses:**

**Limited empirical improvement**
The claimed benefit of the proposed method is not convincingly demonstrated. On most benchmarks, performance on the retain tasks (SQuAD, GSM8K, MathQA) is similar to or only slightly better than standard supervised fine-tuning, while the model loses almost all other general capabilities. The reported gains (e.g., +1–2%) are marginal and could be within the noise level.

**Severe loss of generalization and transferability**
By design, the method intentionally erases general knowledge. However, in many real applications — especially those requiring lifelong learning, domain adaptation, or multi-skill reasoning — prior general knowledge is often essential for solving domain-specific tasks. Erasing all other abilities seems counterproductive and undermines the benefit of large-scale pretraining.

**Questionable motivation and practicality**
The motivation for forgetting “everything else” is not fully justified. In most deployment scenarios (e.g., chatbots, education, vertical assistants), controlling responses via task prompts, instruction-tuning, or safety filters is far more efficient and reversible than permanently unlearning. The proposed approach sacrifices flexibility for very narrow gains.

**Lack of qualitative or downstream analysis**
The paper does not explore what exactly is being forgotten, nor whether the remaining ability maintains robustness or compositional reasoning. This makes it unclear whether “exclusive unlearning” leads to a stable or brittle model.

**Questions:**

How does the proposed approach behave in lifelong learning or multi-task scenarios?
If we later wish to add another ability (e.g., after forgetting everything else), can the model be retrained efficiently, or must it be fully retrained from scratch?

---

### Official Review · Reviewer_GZAH · 2025-11-01

**Soundness:** 2
**Presentation:** 3
**Contribution:** 2
**Rating:** 2
**Confidence:** 3

**Summary:**

The paper introduces a new unlearning paradigm called exclusive unlearning, where the goal is to retain the model’s ability on a specific target task while removing general or unrelated knowledge. To achieve this, the authors propose a method that minimizes a loss combining the target-task objective with an additional loss designed to unlearn the model’s general capabilities. Empirical evaluations show that the proposed approach successfully maintains strong performance on the target task while substantially reducing accuracy on MMLU, indicating effective removal of general knowledge.

**Strengths:**

1. The proposed setting of unlearning, exclusive unlearning, is novel.
2. The writing is structured well and overall it is easy to follow.

**Weaknesses:**

1. Motivation of exclusive unlearning is unclear. While the goal of retaining only the target task capability is understandable, it is not evident why removing unrelated abilities—such as those measured by MMLU—would be beneficial. In most practical scenarios, preserving general capabilities seems harmless. A stronger justification for why “exclusive unlearning” is desirable in real-world applications would make the contribution more compelling.
2. If the paper would like to justify all other abilities can be unlearned, then the set-up of empirical evaluation is insufficient -- only MMLU is evaluated. There are many other benchmarks evaluating model's general ability (PIQA, RACE) as well as the fact datasets such as Wiki that can facilitate to measure how much factual knowledge is in the LLM.
3. Another baseline can be learning the target dataset from randomly initialized models rather than the pre-trained models.

**Questions:**

Please check the weaknesses section.

---

### Official Review · Reviewer_55Xo · 2025-11-02

**Soundness:** 3
**Presentation:** 4
**Contribution:** 3
**Rating:** 8
**Confidence:** 3

**Summary:**

This paper describes a new method for selectively unlearning in LLMs. Unlike prior work which focuses on erasing specific knowledge while preserving other broader abilities, the authors propose exclusive unlearning, i.e., preserving only a designated ability while forgetting all other knowledge and capabilities. The method fine-tunes the model on a target task (retain set) while simultaneously applying a “uniform-loss” objective on self-generated text to drive the model’s output distribution toward uniformity (forget set). They evaluate this on extractive QA (SQuAD) and mathematical QA (GSM8K, MathQA) across several model sizes, showing the model retains the target task performance while performance on a broad benchmark (MMLU) drops to near chance.

**Strengths:**

1. The inversion of standard unlearning (i.e., keeping everything but forgetting specific knowledge) into only keeping the specific knowledge and forgetting everything else is a novel conceptual shift.
2. The proposed method using uniform-loss objective is elegant and theoretically sound.
3. Strong empirical results across model scales and tasks. The author showed the effectiveness of the proposed method on multiple model families and on two different target tasks (extractive QA and math QA) with convincing drops on the broad task benchmark MMLU.

**Weaknesses:**

1. While the proposed method on using Uni CE showed advantages over Neg CE in the experiments, limited insight was revealed and discussed on why Uni CE works better.
2. While the method fits well for constrained tasks, the trade-offs are under-explored, e.g. what gets lost (e.g. will the language capability also lose in addition to the general knowledge), and what might inadvertently remain.
3. The retain tasks are both QA (extractive and math). It’s unclear how the method generalizes to other modalities (generation, summarization, dialogue) or less structured tasks.

**Questions:**

1. What makes Uni CE works so much better than Neg CE in this unlearning setting?
2. How sensitive is the method to the quality/diversity of the self-generated “forget” dataset, what happens if key domains are under-sampled?
3. Figure 2: Why a large lambda (weight for the retain loss) lead to a drop in performance of the retain task and improvement on forgetting task?

---

### Official Review · Reviewer_Qsxo · 2025-11-08

**Soundness:** 2
**Presentation:** 3
**Contribution:** 2
**Rating:** 2
**Confidence:** 3

**Summary:**

This paper proposes an unlearning method that aims to maintain only necessary knowledge and forget everything else. It achieves this by generating a forget dataset consisting of texts generated by the model itself and pushes the model’s output probability to be as uniform as possible. Meanwhile, it assumes a target task dataset exists to retain information necessary for the task.

**Strengths:**

The method is very straightforward.

The writing is easy to follow.

The motivation of preventing misuse is generally valid.

**Weaknesses:**

The technical approach proposed is not very novel, and too naive to be meaningful. The evaluations are too weak to support the usefulness of the approach.

1. The proposed method that tries to uniformalize the output distribution is not novel, as it has been proven to work for other unlearning and/or debiasing tasks. However, directly using this technique in the vocabulary space is potentially catastrophic, because it not only erases knowledge, but also the general ability of articulation the model obtains through trains.

2. How much the model forgets depends on the size of the forget dataset. If the dataset is too small, the information it forgets is too little; if the dataset is too large, then the model loses the general ability to talk. The paper does not discuss a principled approach to find the correct size of the forget dataset.

3. There is a risk of catastrophically breaking the talking ability of the model and the method does not provide any measure to prevent this from happening. The model only retains the ability to speak on the target dataset. So what would happen if the user inputs any text outside the distribution of the retain/target dataset? - assuming the forget dataset is sufficiently large, the model would essentially output random tokens, instead of rationally rejecting the question, which significantly reduces the usefulness of the model.

4. The paper provides no evidence using a forget dataset and a retain dataset is better than simply training the LLM only on the retain dataset from scratch - what is the benefit of using a pretrained LLM at all if the goal is to literally forget everything not in the retain dataset? If the objective proposed by this paper is to make any output distribution other than

5. The paper provides no evaluations on data outside of the forget dataset and the retain dataset. It is possible the model still remembers the information as long as prompted in a different way from the ones in the forget dataset. It is also possible it becomes very easy to make it output complete random tokens.

**Questions:**

Please see Weaknesses section.

---

> ### Author Response · Authors · 2025-11-20
> **Clarification on the Novelty of Output-Uniformization Methods**
>
> Thank you very much for your constructive review.
>
> We would like to clarify the following weak point:
>
> > The proposed method that tries to uniformalize the output distribution is not novel, as it has been proven to work for other unlearning and/or debiasing tasks.
>
> If such techniques indeed exist in the prior literature, could you kindly point to representative works or commonly cited approaches?
>
> Thank you for your consideration.

---

### Meta-Review · Area_Chair_V5xM · 2026-01-05

**Summary:**

Reviewers consistently criticize the work for limited novelty, weak motivation, forgetting of other knowledge, and insufficient empirical validation.

**Reviewer Concerns:**

Reviewers consistently criticize the work for limited novelty, weak motivation, forgetting of other knowledge, and insufficient empirical validation. The proposed unlearning approach—uniformizing output distributions—is viewed as a naive reuse of existing debiasing techniques and potentially dangerous when applied in vocabulary space, as it risks destroying general language ability. A major concern is the lack of principled guidance on selecting the forget dataset size and the absence of safeguards against catastrophic loss of articulation and generalization. Reviewers question the practicality of “exclusive unlearning,” arguing that erasing all non-target capabilities is poorly motivated and often counterproductive compared to simpler alternatives such as training from scratch or using prompting and fine-tuning. Empirical evaluations are seen as narrow, relying on limited tasks and benchmarks, with marginal gains and no analysis of out-of-distribution behavior, residual knowledge, or transferability.

**Reviewer Scores:**

The authors did not submit a rebuttal.

---

### Decision · Program_Chairs · 2026-01-26

Reject